# Seasonal Shifts in Influenza, Respiratory Syncytial Virus, and Other Respiratory Viruses After the COVID-19 Pandemic: An Eight-Year Retrospective Study in Jalisco, Mexico

**DOI:** 10.3390/v16121892

**Published:** 2024-12-08

**Authors:** Ernestina Quintero-Salgado, Jaime Briseno-Ramírez, Gabriel Vega-Cornejo, Roberto Damian-Negrete, Gustavo Rosales-Chavez, Judith Carolina De Arcos-Jiménez

**Affiliations:** 1State Public Health Laboratory, Zapopan 45170, Jalisco, Mexico; titinaquintero2@gmail.com; 2Health Division, Tlajomulco University Center, University of Guadalajara, Tlajomulco de Zuñiga 45641, Jalisco, Mexico; jaime.briseno@academicos.udg.mx (J.B.-R.); gabriel.vega@academicos.udg.mx (G.V.-C.); roberto.damian@cutlajomulco.udg.mx (R.D.-N.); grosales@hcg.gob.mx (G.R.-C.); 3Antiguo Hospital Civil de Guadalajara “Fray Antonio Alcalde”, Guadalajara 44280, Jalisco, Mexico; 4Hospital General de Occidente, Zapopan 45170, Jalisco, Mexico; 5Laboratory of Microbiological, Molecular and Biochemical Diagnostics (LaDiMMB), CUTlajomulco, University of Guadalajara, Tlajomulco de Zuñiga 45641, Jalisco, Mexico; 6Nuevo Hospital Civil de Guadalajara “Dr. Juan I. Menchaca”, Guadalajara 4340, Jalisco, Mexico

**Keywords:** respiratory viruses’ trends, COVID-19 impact, virus seasonality, respiratory infections, influenza seasonality, virus resurgence, interrupted time series analysis

## Abstract

The coronavirus disease 2019 (COVID-19) pandemic profoundly disrupted the epidemiology of respiratory viruses, driven primarily by widespread non-pharmaceutical interventions (NPIs) such as social distancing and masking. This eight-year retrospective study examines the seasonal patterns and incidence of influenza virus, respiratory syncytial virus (RSV), and other respiratory viruses across pre-pandemic, pandemic, and post-pandemic phases in Jalisco, Mexico. Weekly case counts were analyzed using an interrupted time series (ITS) model, segmenting the timeline into these three distinct phases. Significant reductions in respiratory virus circulation were observed during the pandemic, followed by atypical resurgences as NPIs were relaxed. Influenza displayed alternating subtype dominance, with influenza A H3 prevailing in 2022, influenza B surging in 2023, and influenza A H1N1 increasing thereafter, reflecting potential immunity gaps. RSV activity was marked by earlier onset and higher intensity post-pandemic. Other viruses, including human rhinovirus/enterovirus (HRV/HEV) and parainfluenza virus (HPIV), showed altered dynamics, with some failing to return to pre-pandemic seasonality. These findings underscore the need for adaptive surveillance systems and vaccination strategies to address evolving viral patterns. Future research should investigate the long-term public health implications, focusing on vaccination, clinical outcomes, and healthcare preparedness.

## 1. Introduction

Influenza and respiratory syncytial virus (RSV) are well known for causing substantial respiratory illnesses, particularly among infants and older adults [1,2]. The COVID-19 pandemic significantly altered the transmission patterns of various respiratory viruses, including influenza virus, RSV, and other seasonal pathogens [3,4].

Before the pandemic, respiratory viruses typically exhibited predictable seasonal patterns, with peaks often occurring during the winter months in temperate regions [5,6]. However, the COVID-19 pandemic disrupted this predictability, likely due to a combination of multiple factors, including widespread non-pharmaceutical interventions (NPIs) such as masking and social distancing [7,8], immune and viral interference [9,10], changes in population susceptibility [11], genetic adaptations and bottlenecking [12,13], increased vaccination efforts [14,15], and shifts in human behavior [16]. During the 2020–2021 influenza and RSV season, circulation levels dropped to historically low levels, and the usual winter epidemics failed to materialize in many regions of the Northern and Southern Hemispheres [17,18,19]. Similar reductions and altered seasonal patterns were observed globally, including in Europe, Australia, New Zealand, and other regions [20,21].

As NPIs were relaxed, respiratory viruses, including influenza viruses and RSV, gradually returned to pre-pandemic circulation levels, often outside their traditional seasons. For instance, in the Northern Hemisphere, the 2021–2022 season saw earlier RSV peaks compared with pre-pandemic norms [21,22]. Off-season increases were particularly evident in 2021, especially among children and young adults, coinciding with the relaxation of public health measures [20,22]. In China, a year-round RSV outbreak was reported in 2021, with heightened detection rates and a lack of the typical seasonal pattern observed in prior years [23,24]. The 2022–2023 RSV season witnessed a significant resurgence of cases, nearing pre-pandemic incidence levels [25].

The resurgence of influenza following COVID-19 has been complex, with regional differences and changes in population immunity playing key roles. For example, in 2021, influenza A resurged in the United States, whereas influenza B predominated in China, highlighting regional variability in resurgence patterns [26]. In South Korea, the incidence of influenza returned to pre-pandemic levels during the 2022–2023 season, following the relaxation of COVID-19 prevention policies [27]. Similarly, in Victoria, Australia, influenza re-emerged after travel restrictions were lifted, with young adults and international travelers contributing significantly to transmission dynamics [28]. In the United States, the 2021–2022 influenza season was associated with an increased risk of household transmission of influenza A (H3N2), likely due to reduced population immunity [29]. In China, during the 2022–2023 season, a notable increase in influenza cases and susceptibility was observed, attributed to the lifting of COVID-19 restrictions and the resulting “immunity debt” [30,31].

This resurgence of influenza, RSV, and other respiratory viruses in the post-pandemic era has raised significant public health concerns [30]. Following the influenza A (H1N1) pandemic in 2009, the seasonality and age distribution of other respiratory viruses changed, suggesting that pandemics can have lasting effects on the epidemiology of these pathogens [32]. The unpredictability of seasonal patterns complicates traditional models of disease prediction and management [33].

This study aims to describe the incidence, trends, and seasonality of influenza, RSV, and other respiratory viruses before and after the COVID-19 pandemic. The goal is to enhance surveillance and diagnostics, which are essential for guiding effective public health responses to these respiratory pathogens.

## 2. Materials and Methods

### 2.1. Ethics

This study involving human participants was reviewed by the Research Committee of the Ministry of Health of Jalisco and approved for inclusion in the State Research Registry under number 73/LESP/JAL/2024. Ethical approval was granted by the “Comité de Ética en Investigación de la Secretaría de Salud de Jalisco” (approval number SSJ/DGEICS/DIS/CEI/12/24), and by the Research Committee (approval number SSJ/DGEICS/DIS/CI/13/24). This research was conducted in accordance with the principles of the Declaration of Helsinki, applicable national legislation, and institutional guidelines. Due to the retrospective nature of this study and the exclusive use of de-identified data, informed consent was waived.

### 2.2. Population and Eligibility Criteria

We retrospectively reviewed the state laboratory registry of symptomatic patients with influenza-like illness (ILI) tested for respiratory viruses from January 2017 to October 2024. A suspected case of ILI was defined as the sudden onset of symptoms accompanied by at least one of the following systemic symptoms: fever or feverishness, cough, or headache; and at least one of the following localized symptoms: dyspnea, myalgias, arthralgias, odynophagia, chills, chest pain, rhinorrhea, tachypnea, anosmia, dysgeusia, or conjunctivitis [34]. Demographic data and the presence of comorbidities were systematically collected from patient records. Cases with more than 10% missing data in sociodemographic or clinical records were excluded from the analysis.

### 2.3. Viral Testing

The respiratory viruses of interest included SARS-CoV-2, influenza A, influenza B, respiratory syncytial virus (RSV), human parainfluenza viruses 1–4 (HPIV1, HPIV2, HPIV3, HPIV4), human metapneumovirus (hMPV), seasonal human coronaviruses (HCoV-229E, HCoV-OC43, HCoV-NL63, and HCoV-HKU1), human adenovirus (HAdV), human bocavirus (HBoV), and human enterovirus/rhinovirus (HEV/HRV). Viral detection was performed using Health Mexico-approved assays with single or multiplex reverse transcriptase real-time PCR (RT–PCR) assays. A case was defined as a laboratory-confirmed positive test for any of the respiratory viruses. Nasopharyngeal swabs were collected in viral transport media and transported to the State Laboratory under cold chain conditions. Upon arrival, laboratory procedures included viral inactivation, nucleic acid extraction, and viral gene amplification using RT–PCR.

### 2.4. Testing Restrictions

According to local guidelines prior to the COVID-19 pandemic, respiratory viruses’ surveillance primarily focused on influenza but improved significantly with the onset of COVID-19. Before the pandemic, only 10% of outpatients with suspected influenza and all hospitalized patients with suspected influenza were tested for influenza A and B. Additionally, 10% of patients who tested negative for influenza were screened for other respiratory viruses [35]. Following the onset of the COVID-19 pandemic, 100% of suspected cases were tested for SARS-CoV-2 and influenza A and B using multiplex RT–PCR. Other respiratory viruses were tested in 10% of these cases [36].

### 2.5. Statistical Analysis

Demographic data were reported as simple relative frequencies. The percentage of positive test results was calculated by dividing the number of positive tests by the total number of tests conducted during a specified period, expressed as a percentage for each virus. The normality of the data distribution was evaluated using the Shapiro–Wilk test. To compare proportions, Pearson’s chi-square test and Fisher’s exact test were employed as appropriate. Quantitative variables were compared using Student’s *t*-test for normally distributed data and the Wilcoxon–Mann–Whitney test for non-normally distributed data.

To assess the impact of the COVID-19 pandemic on respiratory viruses’ circulation, an interrupted time series (ITS) analysis was performed on weekly case numbers for multiple respiratory viruses. A generalized linear model with a negative binomial distribution was used to address overdispersion, enabling the quantification of both immediate and gradual effects of pandemic interventions on viral activity. The timeline was divided into three distinct phases: the pre-pandemic phase (January 2017–March 2020), the pandemic phase (April 2020–December 2021), and the post-pandemic phase (January 2022 onward). The pre-pandemic period served as the baseline, while the pandemic phase captured the initial effects of strict lockdowns and subsequent public health measures. The post-pandemic phase reflected the changes observed following the relaxation of these interventions.

The ITS model was designed to estimate both abrupt changes in level and gradual changes in slope, effectively capturing dynamic shifts in viral incidence over time. Seasonal trends were accounted for by analyzing data on a weekly basis, allowing the model to reflect typical fluctuations in respiratory viruses’ activity. Sensitivity analyses were performed using three distinct models to ensure the robustness of the findings. The first model assessed the impact of strict lockdown periods by examining level changes and trends during the early stages of the pandemic. The second model focused on seasonal effects, analyzing intra-year fluctuations to gain a more detailed understanding of variations in viral circulation. The third model incorporated weekly COVID-19 case counts to evaluate their concurrent influence on the incidence of other respiratory viruses. A significance level of 5% (*p* < 0.05) was applied to all statistical tests to identify meaningful changes in respiratory viruses’ activity across the different pandemic phases.

All statistical analyses were conducted using Python (version 3.12). Data management was performed using the Pandas library (version 1.5.0), while interrupted time series regression models were implemented with the Statsmodels library (version 0.14.0). Statistical computations were conducted using the SciPy library (version 1.11.0). To address overdispersion, the ITS models were constructed using negative binomial regression. The variance inflation factor (VIF) was calculated using the Statsmodels package to assess multicollinearity.

### 2.6. Data Visualiztion

To visualize the incidence and positivity rates of respiratory viruses (excluding SARS-CoV-2) and assess the potential impact of the COVID-19 pandemic on their seasonality and trends, a series of graphs were developed. These visualizations compared laboratory data with state-level metrics obtained from open-access datasets provided by the Federal Ministry of Health and the State Ministry of Health [37,38]. The metrics included the total number of COVID-19 cases, total cases of acute respiratory illness (regardless of etiology), COVID-19-related hospitalizations, hospitalizations due to acute respiratory illness, COVID-19-related deaths, and deaths from acute respiratory illness, all specific to the state of Jalisco. These datasets incorporated results from both rapid antigen tests and RT-PCR assays, ensuring comprehensive diagnostic coverage. In addition, open data on the prevalence of predominant SARS-CoV-2 variants in Mexico were integrated into the graphs to construct a timeline of variant predominance [39]. This allowed for the assessment of potential associations between variant prevalence, the incidence of other respiratory viruses, and key metrics such as hospitalizations and deaths.

To evaluate the potential impact of vaccination campaigns on respiratory viruses’ trends, the timeline of the sequential COVID-19 vaccination rollout in Mexico, including booster dose distribution, was integrated into the visualizations [40]. This timeline detailed the rollout by priority groups (e.g., healthcare workers, age groups, and vulnerable populations) and was overlaid on the graphs to examine temporal relationships between vaccination milestones, respiratory viruses’ trends, and public health outcomes. Finally, the timeline of NPIs implemented by the Government of the State of Jalisco was included. These were categorized as strict (e.g., stay-at-home orders, closure of non-essential businesses, suspension of public transport), moderate (e.g., capacity limits in public venues, mandatory mask use, localized restrictions in hotspots), dynamic measures (adjustments to restrictions based on viral surges), and mitigation measures (e.g., encouragement of vaccination, use of face masks, physical distancing, and localized testing). This facilitated the evaluation of the impact of both NPIs and vaccination campaigns on the incidence and seasonality of respiratory viruses during the pandemic and post-pandemic periods.

## 3. Results

A total of 157,418 subjects were identified from laboratory registries, of whom 155,352 met the inclusion criteria. Among these, 57,507 tests were conducted for respiratory viruses other than SARS-CoV-2. This included 57,389 influenza tests and 17,253 tests for respiratory viruses distinct from SARS-CoV-2 and influenza, as shown in Figure 1.

The median age of individuals tested was 37 years (IQR 25–53), with 53.97% being female (*n* = 83,841). After excluding those positive for SARS-CoV-2, the median age of tested individuals decreased to 34 years (IQR 18–53), while the median age of subjects with laboratory-confirmed viral respiratory infections (n = 9889) further declined to 22 years (IQR 2–38), with females accounting for 55.0% (n = 5439). Table 1 presents descriptive statistics of respiratory virus cases before and after the onset of the COVID-19 pandemic, categorized by sex and age group. Notably, the age distribution shifted significantly toward the 15–65 age group, which accounted for 34.92% of cases (n = 784) before the pandemic and 51.27% (n = 3919) after (*p* < 0.001). A detailed breakdown by respiratory virus type is provided in Appendix A.

Comorbidities were present in 31.14% of the confirmed cases (n = 3079). The most frequently documented comorbidities were hypertension (n = 817), obesity (n = 660), smoking (n = 647), diabetes (n = 613), and asthma (n = 640). Additional sociodemographic data are summarized in Table 2.

Among the confirmed cases of respiratory viruses, 55,153 were SARS-CoV-2, 5325 were influenza virus, and 3958 were other respiratory viruses distinct from SARS-CoV-2 and influenza. Regarding influenza subtypes, influenza A H3 was the most prevalent (30.02%, n = 2969), followed by influenza B Victoria lineage (15.28%, n = 1511) and influenza A H1N1 (3.30%, n = 326). Among respiratory viruses, RSV was the most frequently identified (15.56%, n = 1539), followed by HEV/HRV (14.81%, n = 1465), hMPV (14.92, n = 487), and HPIV (3.80%, n = 376). Additional details on the remaining detected respiratory viruses are provided in Table 3.

A total of 989 coinfections were documented, with the most frequent combinations being SARS-CoV-2 and influenza (28%, n = 277), HEV/HRV and RSV (17.79%, n = 176), HAdV and HEV/HRV (8.59%, n = 85), SARS-CoV-2 and HEV/HRV (6.47%, n = 64), HAdV and RSV (6.26%, n = 62), HPIV and HEV/HRV (6.06%, n = 60), and SARS-CoV-2 and RSV (5.66%, n = 56) (Figure 2a).

Among 11,909 tests conducted using the multiplex RT-PCR respiratory panel, which detects COVID-19 and other respiratory viruses, 9.67% (n = 1152) were positive for SARS-CoV-2. Of these, 12.07% (n = 139) were identified as coinfections with other respiratory viruses. The most frequent coinfections in this group were SARS-CoV-2 with HEV/HRV (5.55%, n = 64), SARS-CoV-2 with RSV (4.86%, n = 56), and SARS-CoV-2 with HAdV (2.43%, n = 28). Other coinfections documented across the entire study period, and those identified through the multiplex RT-PCR respiratory panel, are presented in Figure 2a,b, respectively. An extended version, including influenza and parainfluenza virus subtypes from the entire study period, is available in Appendix A.

Seasonal patterns were observed for laboratory-confirmed cases of influenza and RSV among patients from late autumn to early spring during the 2016–2017 to 2018–2019 seasons. The end of the 2019–2020 season coincided with the arrival of SARS-CoV-2 in March 2020, when non-pharmaceutical interventions (NPIs) were implemented. These measures led to historically low detection rates, with only one influenza case reported in April 2021 (positivity rate: 0.76%), compared with pre-pandemic influenza winter seasons (mean positivity rate: 12.63%), as illustrated in Figure 3a. 

With the gradual relaxation of NPIs in late 2021, the phased rollout of vaccination, and the emergence of the Omicron variant of SARS-CoV-2, a surge in influenza cases was observed. This surge was accompanied by an out-of-season year-round detection pattern during 2022 and 2023 (mean positivity rate: 13.01% from December 2021 to July 2023), including a peak positivity rate of 39.80% in November 2022, coinciding with three Omicron waves in the population. However, hospitalizations and deaths due to acute respiratory illness during this period were lower than those observed in 2020 and 2021. The incidence of influenza and its subtypes, and their relationship with SARS-CoV-2 incidence, hospitalizations, deaths, NPIs, and the local vaccination strategy, are shown in Figure 3a,b, respectively.

Influenza detection persisted throughout 2022 but decreased in frequency by July 2023, remaining at low levels (positivity rate: 2–5%) until the winter season of 2023–2024, which saw a peak positivity rate of 21%. Interestingly, alternating patterns among influenza subtypes were observed during the two years of persistent detection (2022 and 2023) following the heightened NPI period. Influenza A H3 was the predominant subtype in 2022, followed by a surge in influenza B Victoria lineage cases in February 2023 and a subsequent increase in influenza A H1 in August 2023. Influenza activity declined further by July 2024. Notably, no influenza B Yamagata lineage cases were detected from July 2019 until the time of this report. Detection frequencies and positive test numbers for influenza virus subtypes, along with their relationship to the implementation of NPIs and the COVID-19 vaccination strategy, are presented in Figure 3b.

For RSV, an early onset of activity was observed in August 2021, whereas in previous seasons (2017–2018 to 2019–2020), the season typically began in late October or early November. The early onset of RSV in August 2021, along with its subsequent extension until April 2022, was accompanied by an increase in cases and higher positivity rates compared with the winter seasons of 2017–2018 to 2019–2020. This surge coincided with two waves of SARS-CoV-2 driven by the Delta and Omicron variants during the winter of 2021–2022, and an increase in hospitalizations and deaths due to acute respiratory illness, although these were at a lower proportion than those observed between January 2020 and November 2021. After April 2022, RSV activity declined, returning to its usual seasonal pattern during the 2022–2023 and 2023–2024 seasons. The incidence of RSV, and its relationship with SARS-CoV-2 incidence, hospitalizations, deaths, NPIs, and the local vaccination strategy, are illustrated in Figure 4a,b.

The analysis of respiratory viruses distinct from influenza and RSV revealed a dynamic interplay between these pathogens, SARS-CoV-2 variants, public health interventions, and evolving population immunity. From 2017 to early 2020, the seasonal peaks of various respiratory viruses followed relatively predictable patterns, with sharp increases during the colder months. However, the implementation of stringent NPIs in early 2020, including social distancing, mask wearing, and school closures, resulted in a significant suppression of group positivity rates for respiratory viruses such as HEV/HRV, hMPV, and HPIV. This suppression became evident from May 2020 onwards, with group positivity rates declining markedly and remaining at low levels until mid-2021. Despite this overall decrease, HEV/HRV continued to be detected throughout 2020, coinciding with SARS-CoV-2 waves and the enforcement of strict and moderate NPIs.

By August 2021, as NPIs were gradually relaxed, an early re-emergence of several respiratory viruses was observed. HEV/HRV, hMPV, and HPIV demonstrated early increases in activity, even preceding the typical winter season. Unlike the pre-pandemic period, their resurgence did not follow the established seasonal peaks, but instead showed continuous detection throughout 2021 and 2022. Notably, the circulation of these viruses also coincided with the emergence of the Delta and Omicron variants of SARS-CoV-2. 

For HBoV, HCoV-229E, HCoV-HKU1, HCoV-NL63, and HCoV-OC43, detection frequencies remained consistently low throughout the observation period, making it difficult to identify clear seasonal patterns. The incidence of other respiratory viruses distinct from influenza and RSV, along with their relationship to SARS-CoV-2 incidence, hospitalizations, deaths, NPIs, and the local vaccination strategy, are illustrated in Figure 5a,b. Individual virus detection frequencies are detailed in Appendix A.

The segmented interrupted time series (ITS) analysis for respiratory viruses is summarized in Table 4. During the 2020–2021 period, significant decreases were observed in the levels of influenza (β_2_ = −4.264, *p* < 0.001), RSV (β_2_ = −6.421, *p* < 0.001), HEV/HRV (β_2_ = −1.171, *p* < 0.001), HPIV (β_2_ = −5.910, *p* < 0.001), and HAdV (β_2_ = −2.268, *p* < 0.001). These results highlight a substantial reduction in the number of cases of all analyzed respiratory viruses during the initial years of the COVID-19 pandemic. hMPV also exhibited a significant decline during this period (β_2_ = −1.200, *p* = 0.001), although the magnitude of reduction was smaller compared with the other viruses.

In contrast, during 2022, significant increases were observed for influenza (β_2_ = 9.418, *p* < 0.001), RSV (β_2_ = 5.367, *p* < 0.001), and HPIV (β_2_ = 4.720, *p* < 0.001), suggesting a resurgence of these pathogens following the relaxation of pandemic-related public health interventions. Interestingly, hMPV did not exhibit a significant change during this period (β_2_ = 0.679, *p* = 0.112), while HAdV slightly but not significantly decreased (β_2_ = −0.331, *p* = 0.561). HEV/HRV continued to show a slight but significant decline (β_2_ = −1.250, *p* = 0.002), indicating a different post-pandemic recovery pattern compared with the other viruses.

Trend dynamics across the analyzed periods revealed additional insights. During the 2020–2021 period, significant positive trends were observed for influenza (β_3_ = 0.066, *p* < 0.001), RSV (β_3_ = 0.086, *p* < 0.001), HPIV (β_3_ = 0.056, *p* < 0.001), and hMPV (β_3_ = 0.011, *p* = 0.043), reflecting a gradual recovery in activity. In contrast, HEV/HRV displayed a slight but significant negative trend (β_3_ = −0.016, *p* = 0.002), while HAdV showed no significant trend (β_3_ = −0.006, *p* = 0.412), suggesting stable incidence during this period.

During 2022, trends varied further. Influenza exhibited a slight but significant negative trend (β_3_ = −0.013, *p* < 0.001), as did RSV (β_3_ = −0.024, *p* < 0.001) and HEV/HRV (β_3_ = −0.044, *p* < 0.001). HPIV continued to show a significant negative trend (β_3_ = −0.017, *p* < 0.001), while hMPV also demonstrated a negative trend (β_3_ = −0.016, *p* < 0.001). HAdV followed a similar pattern, with a notable negative trend during this period (β_3_ = −0.043, *p* < 0.001). The complete ITS model of respiratory viruses’ circulation before, during, and after the COVID-19 pandemic is provided in Appendix A.

Sensitivity analyses were conducted to ensure the robustness of the ITS models, with findings summarized in Appendix A. These analyses evaluated three distinct models, each providing additional insights into the dynamics of respiratory viruses’ activity during the COVID-19 pandemic. Model 1, which included the strict lockdown period, revealed significant reductions in virus circulation for most pathogens, including influenza (β_2_ for 2020–2021 = −19.809, *p* < 0.001) and RSV (β_2_ for 2020–2021 = −16.821, *p* < 0.001). These findings underscore the substantial suppression of viral transmission due to pandemic-related public health measures. Similar reductions were observed for hMPV and HPIV during the lockdown phase, with β_2_ values of −14.700 (*p* < 0.001) and −5.784 (*p* < 0.001), respectively. In contrast, Model 2, which aimed to evaluate seasonal patterns, could not be computed for certain viruses, such as influenza and RSV, due to the limited number of cases during the 2020–2021 season, resulting in insufficient statistical power. Model 3, which incorporated weekly COVID-19 case counts, offered additional insights by demonstrating that the resurgence of certain viruses, including influenza (β_2_ for 2022 = 8.954, *p* < 0.001) and RSV (β_2_ for 2022 = 5.438, *p* < 0.001), coincided with fluctuations in COVID-19 cases. For other viruses, such as HEV/HRV and HAdV, distinct dynamics were noted, with varying trends observed during and after the pandemic phases. These findings highlight the nuanced interplay between pandemic control measures and viral activity, reflecting the complex and diverse effects of the pandemic on respiratory viruses. The detailed results of these analyses are presented in Appendix A.

## 4. Discussion

This study provides a comprehensive analysis of respiratory viruses’ patterns over an eight-year period in a middle-income Latin American country, emphasizing significant changes in the seasonal dynamics of influenza virus, RSV, and other respiratory viruses following the COVID-19 pandemic. Our findings align with global reports of decreased respiratory viruses’ activity during the pandemic, attributed to NPIs such as social distancing, mask wearing, and restrictions on travel and gatherings [3,4,5,6,7,8,9,10,11,12,13,14,15,16]. These measures led to a marked reduction in cases, supporting the hypothesis that NPIs played a critical role in limiting the spread of respiratory pathogens beyond SARS-CoV-2 [19].

Post-pandemic data, however, reveal a resurgence of influenza and RSV cases, often occurring outside their traditional seasonal patterns [41]. Influenza cases, for instance, spiked during the off-peak season, suggesting a possible recalibration of viral seasonality. Similarly, RSV cases, although primarily within the winter period, exhibited an earlier onset, higher case numbers, and increased positivity rates in 2021, following 19 months of low detection due to limited exposure. These findings raise important questions about the pandemic’s long-term effects on respiratory viruses’ epidemiology and underscore the need for adaptive public health strategies to address these emerging trends.

Our ITS analysis demonstrated distinct patterns across the pre-pandemic, pandemic, and post-pandemic phases. Abrupt reductions in virus incidence were observed at the onset of the pandemic, followed by gradual increases in activity as NPIs were lifted. Notably, while influenza and RSV showed significant post-pandemic rebounds, some viruses, such as hMPV and HAdV, displayed more muted responses, reflecting virus-specific variability in resilience to public health interventions. ITS analysis has proven invaluable for evaluating the impact of interventions, such as the COVID-19 pandemic and associated NPIs, on disease incidence over time [6]. This method quantifies both the immediate and long-term effects of interventions on viral transmission and has been instrumental in understanding the dynamics of viral rebounds following the relaxation of NPIs [42,43]. To our knowledge, this is the first study to apply an ITS analysis to respiratory viruses’ trends over an eight-year period in a middle-income Latin American country.

One notable finding was the alternating dominance of influenza subtypes in the post-pandemic period. Influenza A H3 predominated in 2022, followed by an increase in influenza B cases during 2023 and the re-emergence of influenza A H1N1 later that year. These shifts may reflect declines in population immunity to these subtypes, driven by changes in transmission dynamics during the pandemic. Previous studies have shown similar patterns of alternating dominance among influenza A subtypes, such as A(H1N1) and A(H3N2), across different seasons [44,45]. These subtype-specific dynamics hold critical implications for vaccine development and highlight the importance of continuous surveillance and flexibility in public health planning to address evolving viral threats. Interestingly, no influenza B Yamagata lineage cases have been identified since July 2019, consistent with international reports indicating that this lineage has not been definitively detected since April 2020, raising the possibility of its extinction [46,47].

The observed post-pandemic seasonality and incidence of other respiratory viruses, including human HEV/HRV and HPIV, further reflect the pandemic’s impact on viral ecology. While some viruses returned to pre-pandemic levels, others exhibited altered dynamics, suggesting a complex restructuring of viral ecosystems with implications that warrant further investigation [13,48].

The resurgence of multiple respiratory viruses following the COVID-19 pandemic can be attributed to several factors. The reduced circulation of these viruses during the pandemic, primarily driven by widespread non-pharmaceutical interventions (NPIs), may have led to a decline in natural immunity within the population, particularly among individuals who missed regular exposure to these pathogens [48,49]. This phenomenon, often referred to as “immunity debt”, describes the diminished population immunity resulting from reduced exposure to common pathogens during periods of strict NPIs [48,50]. Consequently, the pool of susceptible individuals likely expanded, and the relaxation of NPIs may have facilitated increased transmission and outbreaks of respiratory infections, such as RSV and influenza, which in some cases exceeded pre-pandemic levels [50,51]. 

Although NPIs significantly reduced the transmission of respiratory viruses other than SARS-CoV-2—evidenced by sharp declines in incidence and associated hospitalizations worldwide, with reductions exceeding 60–85% in various settings—the observed changes in circulation, trends, and seasonality of these viruses may not be solely attributable to NPIs [51,52,53]. In several regions, the epidemiology of non-SARS-CoV-2 respiratory viruses appeared partly independent of NPIs, with certain viruses resuming traditional patterns during different periods, regardless of COVID-19 waves or the reintroduction of NPIs [51,54]. This phenomenon has been particularly documented for HEV/HRV and RSV [15,54]. In our population, although overall positivity rates for respiratory viruses other than SARS-CoV-2 decreased, HEV/HRV continued to be detected throughout 2020, coinciding with SARS-CoV-2 waves and the implementation of strict and moderate NPIs. However, HEV/HRV incidence and positivity rates increased during the winter of 2021–2022, aligning with the Delta and Omicron waves of SARS-CoV-2 and the relaxation of NPIs.

Furthermore, human behavior and changes in testing practices, combined with increased public awareness, may have contributed to the observed shifts, as heightened surveillance identified more infections [55]. During the pandemic, individuals were less likely to seek medical care for mild respiratory symptoms due to fear of contracting COVID-19 or overwhelming healthcare systems [56]. This behavioral change likely resulted in underreporting and reduced transmission of other respiratory viruses during the early stages of the pandemic. Conversely, the COVID-19 pandemic accelerated the development and implementation of advanced technologies for virus detection, including antibody-based tests and genomic surveillance techniques. These advancements have facilitated faster and more accurate detection of viral infections, potentially contributing to an apparent increase in reported cases [52,57,58,59].

The overall reduction in respiratory virus transmission during the pandemic led to a decrease in genetic diversity for several viruses. This reduction may influence the evolutionary dynamics and future behavior of these viruses, potentially resulting in more pronounced outbreaks following the relaxation of NPIs [13]. RSV genetic adaptations during and after the COVID-19 pandemic, particularly in RSV-B, include convergent mutations in the F protein’s antigenic regions. These adaptations could impact viral fitness, immune evasion, and transmission dynamics, potentially altering seasonality, increasing outbreak frequency, and shifting population susceptibility. These findings underscore the importance of ongoing genomic surveillance and adaptive public health strategies to mitigate future risks [60,61].

Another factor that may have contributed is viral exclusion, as reports worldwide have documented reduced coinfection rates for SARS-CoV-2, influenza, and RSV, supporting the hypothesis of a viral exclusionary effect [62]. This phenomenon suggests that these viruses could compete for the same host resources or immune responses, thereby reducing the likelihood of simultaneous infections and potentially leading to an increase in the incidence of other viruses following the decline of SARS-CoV-2 [62,63]. Notably, in our population, when multiplex RT-PCR respiratory panel tests were analyzed, a low proportion of coinfections between SARS-CoV-2 and influenza was observed (0.35%, 4/1152). Similarly, when simplex RT-PCR assays were analyzed, the coinfection rate was 2.44% (273/11,200), supporting the viral exclusion hypothesis between SARS-CoV-2 and influenza. Influenza A virus and RSV can significantly inhibit SARS-CoV-2 replication through robust interferon-mediated innate immune responses. The extent of inhibition was found to depend on the timing of viral infection, with pre-existing viral infections exerting stronger inhibitory effects. This interaction highlights how competition for host immune responses may transiently shape viral transmission patterns [9,64].

Vaccination strategies likely influenced viral interactions during the pandemic. The 2020/21 influenza vaccine reduced SARS-CoV-2 infection rates in Italian healthcare workers through non-specific immune protection mediated by trained immunity [65]. However, this protection diminished significantly with the emergence of the Omicron variant, underscoring the impact of viral evolution on immune escape and reinfection risk [66]. Similarly, influenza vaccines exhibited non-specific benefits, potentially decreasing SARS-CoV-2 susceptibility via immune cross-activation [67]. Despite these protective effects, coinfections with influenza and SARS-CoV-2 have been associated with increased morbidity and mortality [68,69]. The Bacillus Calmette–Guérin (BCG) vaccine exemplifies trained immunity by enhancing innate immune responses to diverse pathogens, including SARS-CoV-2 [67]. While epidemiological studies suggest a correlation between BCG vaccination and reduced COVID-19 mortality, confounding factors may influence these observations. By inducing epigenetic and metabolic reprogramming in immune cells, the BCG vaccine holds promise as a non-specific immunological tool against respiratory infections and future pandemics, pending further clinical validation [70]. 

The intricate interplay of these factors, alongside the direct effects of NPIs, likely played a pivotal role in shaping the epidemiology of respiratory viruses during and after the COVID-19 pandemic. This underscores the importance of further research to deepen our understanding of post-pandemic infectious disease dynamics and their long-term implications.

The limitations of this study primarily stem from its retrospective design. Certain variables were unavailable during data collection and could not be included in the analyses. Additionally, local testing guidelines for respiratory viruses, excluding SARS-CoV-2 and influenza, resulted in significantly fewer tests for other viruses, which may have limited the scope of the findings. Future research should focus on elucidating the mechanisms underlying these shifts in viral seasonality and incidence, including potential changes in host immunity following reduced viral exposure during the pandemic. Further studies should also assess the impact of altered seasonality on clinical outcomes, healthcare burden, and vaccination strategies, particularly for high-risk populations.

## 5. Conclusions

This study highlights significant disruptions and shifts in the seasonal dynamics of respiratory viruses following the COVID-19 pandemic, particularly for influenza and other respiratory viruses. Interrupted time series analysis revealed that the emergence of SARS-CoV-2 led to substantial reductions in respiratory virus circulation, followed by atypical trends. These shifts were likely driven by the implementation and subsequent relaxation of NPIs, along with factors such as declining population immunity, possible viral interference, changes in human behavior, increased public awareness, expanded testing, and other contributing elements. Influenza demonstrated alternating subtype dominance, with A(H3) prevailing in 2022, influenza B surging in 2023, and a subsequent rise in influenza A H1N1 cases, suggesting potential immunity gaps and shifts in population susceptibility. RSV exhibited earlier and more intense seasonal peaks post-pandemic, indicating changes in virus–host dynamics. These findings underscore the complexity of post-pandemic viral ecology and emphasize the need for adaptable public health strategies, robust surveillance systems, and targeted vaccination programs to address evolving viral transmission patterns. Further research is essential to assess the long-term implications of these shifts on disease burden and healthcare systems, particularly among vulnerable populations.

## Figures and Tables

**Figure 1 viruses-16-01892-f001:**
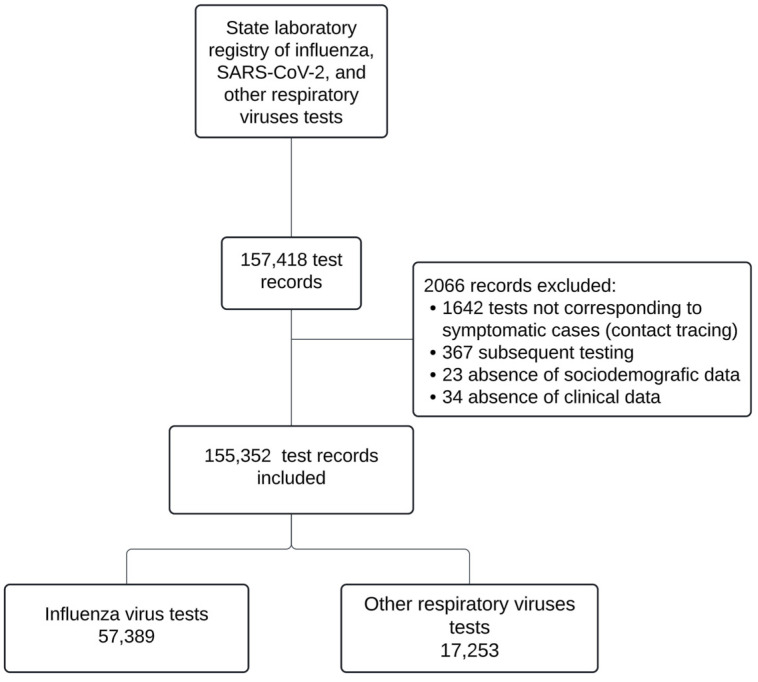
Flowchart of case selection.

**Figure 2 viruses-16-01892-f002:**
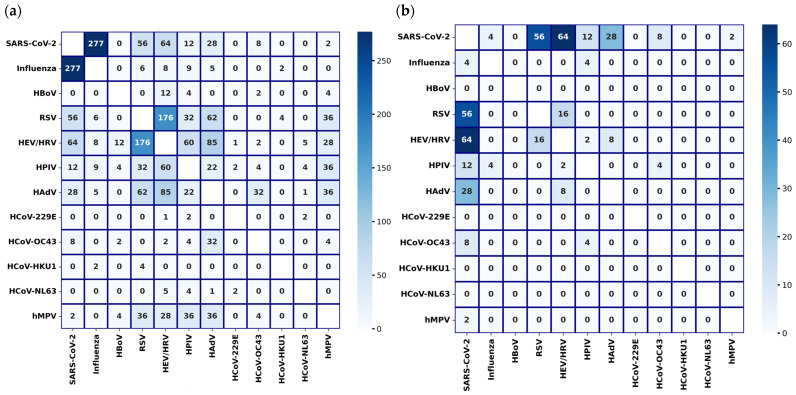
Coinfections detected among different respiratory viruses: (**a**) coinfections detected during the entire study period (2017–2024); (**b**) coinfections detected using the multiplex RT-PCR respiratory panel (2020–2024).

**Figure 3 viruses-16-01892-f003:**
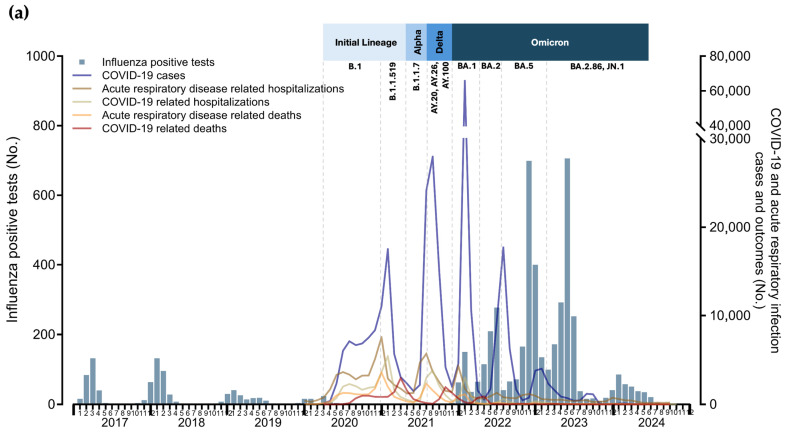
Monthly distribution of positive influenza virus cases: (**a**) monthly distribution of positive influenza cases (bars) alongside COVID-19 indicators and acute respiratory illness metrics (lines); (**b**) monthly distribution of positive cases by influenza subtypes (bars) alongside influenza positivity rates (lines).

**Figure 4 viruses-16-01892-f004:**
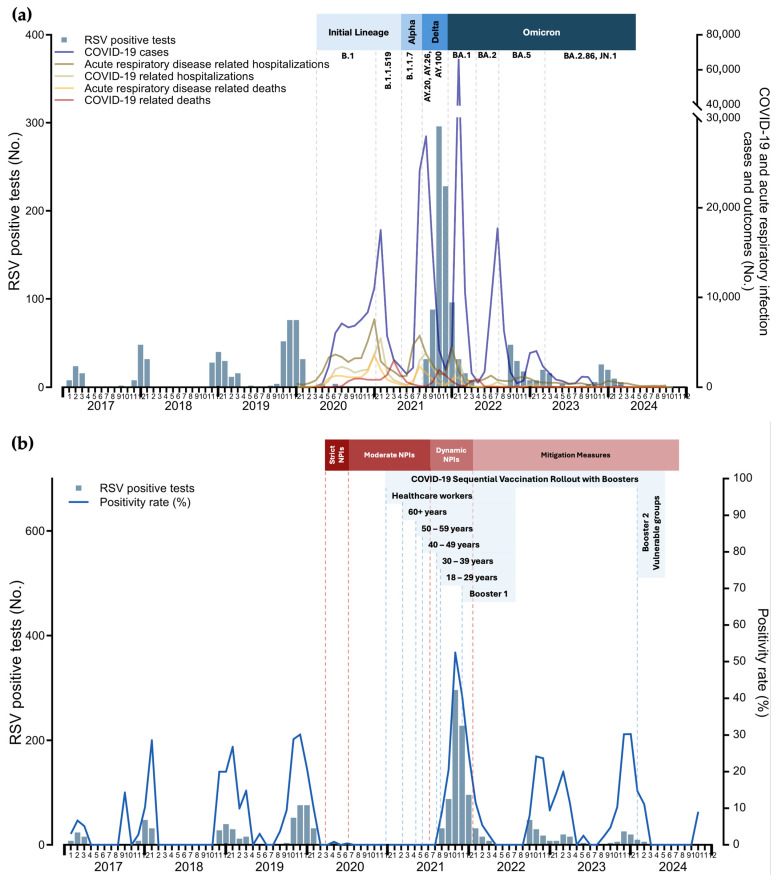
Monthly distribution of respiratory syncytial virus (RSV) positive tests: (**a**) monthly distribution of positive RSV cases (bars) alongside COVID-19 indicators and acute respiratory illness metrics (lines); (**b**) monthly distribution of positive RSV tests (bars) alongside RSV positivity rates (lines).

**Figure 5 viruses-16-01892-f005:**
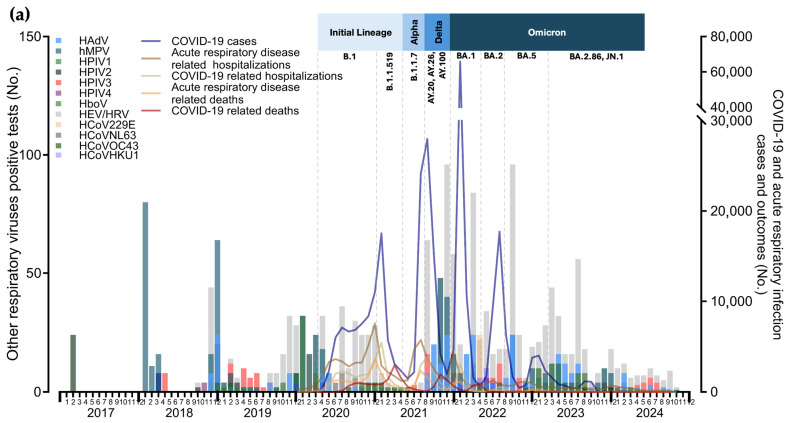
Monthly distribution of positive tests for other respiratory viruses: (**a**) monthly distribution of positive cases by virus type (bars) alongside COVID-19 indicators and acute respiratory illness metrics (lines); (**b**) monthly distribution of positive tests for other respiratory viruses (bars) alongside their group positivity rates (lines).

**Table 1 viruses-16-01892-t001:** Distribution of laboratory-confirmed viral respiratory infection cases before and after the COVID-19 pandemic, categorized by sex and age group.

	Before Pandemic			After Pandemic			*p*-Value
	Cases (n)	(%)	Annual Mean	Cases (n)	(%)	Annual Mean	
Sex							
Female	1200	53.45	300	4239	55.45	847.8	0.098
Male	1045	46.55	261.25	3405	44.55	681	
Age group							
0–2	733	32.65	183.25	1616	21.14	326.2	<0.001
3–5	347	15.46	86.75	653	8.54	130.6	
6–14	164	7.31	41.0	865	11.32	173	
15–65	784	34.92	196	3919	51.27	783.8	
>65	217	9.67	54.25	591	7.73	118.2	

**Table 2 viruses-16-01892-t002:** Sociodemographic and clinical characteristics of patients with detected respiratory viruses.

Variable	Total of Positive Tests (n = 9889)	Total of Positive Tests in Men (n = 4450)	Total of Positive Tests in Women (n = 5439)	*p*-Value
Age—median, (IQR)	22 [2–38]	15 [1–36]	23 [2–38]	<0.001
Comorbidities—n, (%)	3079 (31.14%)	1325 (29.78%)	1754 (32.25%)	0.008
Asthma—n, (%)	640 (6.47%)	230 (5.17%)	410 (7.54%)	<0.001
COPD—n, (%)	397 (4.01%)	160 (3.60%)	237 (4.36%)	0.055
Smoking—n, (%)	647 (6.54%)	397 (8.92%)	250 (4.60%)	<0.001
Diabetes—n, (%)	613 (6.20%)	263 (5.91%)	350 (6.44%)	0.282
Hypertension—n, (%)	817 (8.26%)	353 (7.93%)	464 (8.53%)	0.282
Cardiovascular disease—n, (%)	248 (2.51%)	123 (2.76%)	125 (2.30%)	0.141
Chronic kidney disease—n, (%)	179 (1.81%)	100 (2.25%)	79 (1.45%)	0.003
Obesity—n, (%)	660 (6.67%)	280 (6.29%)	380 (6.99%)	0.169
Immunosuppression—n, (%)	401 (4.06%)	250 (5.62%)	151 (2.78%)	<0.001
HIV infection—n, (%)	88 (0.89%)	60 (1.35%)	28 (0.51%)	<0.001
Pregnancy—n, (%)	188 (1.90%)	-	188 (3.46%)	-

**Table 3 viruses-16-01892-t003:** Detection of respiratory viruses other than SARS-CoV-2 in patients with respiratory symptoms.

Virus	Total of Positive Tests (n = 9889)—n, (%)	Total of Positive Tests in Men (n = 4450)—n, (%)	Total of Positive Tests in Women (n = 5439)—n, (%)	*p*-Value
Influenza virus	5325 (53.85)	2198 (49.39)	3127 (57.49)	<0.001
Influenza A	3678 (37.19)	1528 (34.34)	2150 (39.53)	<0.001
Influenza A H3	2969 (30.02)	1208 (27.15)	1761 (32.38)	<0.001
Influenza A H1N1	326 (3.30)	169 (3.80)	157 (2.89)	0.001
Influenza A non-subtyped	387 (3.91)	153 (3.44)	234 (4.30)	0.032
Influenza B	1647 (16.65)	670 (15.06)	977 (17.96)	<0.001
Influenza B Victoria lineage	1511 (15.28)	611 (13.73)	900 (16.55)	<0.001
Influenza B Yamagata lineage	105 (1.06)	43 (0.97)	62 (01.14)	0.421
Influenza B non-subtyped	32 (0.32)	17 (0.38)	15 (0.28)	0.345
Other respiratory viruses	3968 (40.13)	1967 (44.20)	2001 (36.79)	<0.001
Respiratory syncytial virus	1539 (15.56)	760 (17.08)	779 (14.32)	<0.001
Human enterovirus/rhinovirus	1465 (14.81)	719 (16.16)	746 (13.72)	<0.001
Human metapneumovirus	487 (4.92)	222 (4.99)	265 (4.87)	0.733
Human parainfluenza virus 1	76 (0.77)	48 (1.08)	28 (0.51)	0.001
Human parainfluenza virus 2	86 (0.87)	74 (1.66)	12 (0.22)	<0.001
Human parainfluenza virus 3	168 (1.70)	96 (2.16)	72 (1.32)	<0.001
Human parainfluenza virus 4	46 (0.47)	25 (0.56)	21 (0.39)	0.176
Human adenovirus	366 (1.47)	150 (3.37)	216 (4.87)	0.130
Human coronavirus 229E	71 (3.70)	28 (0.63)	43 (0.79)	0.358
Human coronavirus OC43	109 (1.10)	43 (0.97)	66 (1.21)	0.255
Human coronavirus HKU1	42 (0.42)	21 (0.47)	21 (0.39)	0.468
Human coronavirus NL63	59 (0.60)	32 (0.72)	27 (0.50)	0.145
Human bocavirus	58 (0.59)	36 (0.81)	22 (0.40)	0.008

**Table 4 viruses-16-01892-t004:** Interrupted time series (ITS) analysis of respiratory viruses’ circulation before, during, and after the COVID-19 pandemic in Jalisco, Mexico.

	Influenza			RSV *			HEV/HRV *			hMPV *			HPIV *		
Variable	β	SE	*p*-Value	β	SE	*p*-Value	β	SE	*p*-Value	β	SE	*p*-Value	β	SE	*p*-Value
Constant (β0)	2.365	0.166	<0.001	0.215	0.193	0.265	−2.821	0.360	<0.001	0.016	0.208	0.937	−0.964	0.254	<0.001
Time (β1)	−1.030	0.180	<0.001	0.939	0.191	<0.001	2.988	0.301	0.566	0.405	0.210	0.054	1.049	0.240	<0.001
Level (β2)															
2020–2021	−4.266	0.670	<0.001	−6.428	0.629	<0.001	−1.170	0.308	<0.001	−1.200	0.364	0.001	−5.912	1.035	<0.001
2022	9.418	0.686	<0.001	5.367	0.658	<0.001	−1.250	0.402	0.002	0.679	0.428	0.112	4.720	1.060	<0.001
Trend (β3)															
2020–2021	0.066	0.010	<0.001	0.086	0.009	<0.001	−0.016	0.005	0.002	0.011	0.005	0.043	0.056	0.014	<0.001
2022	−0.013	0.003	<0.001	−0.024	0.003	<0.001	−0.044	0.004	<0.001	−0.016	0.003	<0.001	−0.017	0.004	<0.001

* RSV—respiratory syncytial virus. HEV/HRV—human enterovirus/rhinovirus. HPIV—human parainfluenza virus (HPIV), and hMPV—human metapneumovirus.

## Data Availability

The raw data supporting the conclusions of this article will be made available by the authors on request.

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
