# Peer review of "Seasonal Shifts in Influenza, Respiratory Syncytial Virus, and Other Respiratory Viruses After the COVID-19 Pandemic: An Eight-Year Retrospective Study in Jalisco, Mexico"

_viruses, 2024, doi:10.3390/v16121892_

Round 1

Reviewer 1 Report

Comments and Suggestions for Authors

Dear Authors, Congratulations on a detailed study. May I suggest a few areas for improvement.

1. Line 44. NPIs are not the sole mechanism for changes in pathogen prevalence. It is suggested that you need to research the area of pathogen and virus interference and modify your approach accordingly.

2. COVID-19 presented as a series of variants. Each variant had a unique age profile for deaths which suggests unique age profiles for morbidity as well. 

3. From one of the reviews on pathogen interference you will see evidence for patterns of interference between vaccines and preceding infections. My own unpublished research shows that prior to the Omicron variant prior influenza vaccination provided a protective effect, however, upon the arrival of Omicron this protection dissapeared. BCG vaccination also provided a wider protective effect.

4. Since the study is based in Mexico it would be useful to provide a timeline for the major COVID-19 variants and their prevalence. Match all your data for other viruses against this timeline for Figures 3,4,etc. Perhaps add the number of COVID-19 deaths as a background profile. From memory, one study from Mexico suggested that Influenza returned around the time of Omicron.

5. Comment on Figure 2 in the light of virus interference.

6. Perhaps add some comments regarding potential nonspecific effects of COVID vaccination. Nonspecific effects of Influenza and BCG vaccination were mentioned above.

7. Maybe add some comments on the varied literature regarding the effectiveness of NPIs

I hope that these comments will increase the wider applicability and understanding of the study results. In particular NPIs are not the sole cause of the observed patterns.

Reviewer 2 Report

Comments and Suggestions for Authors

The article makes a favorable impression and is written with clarity and consistency. The materials and methods are thoroughly detailed, and all relevant data regarding compliance with these measures during the study are provided. Additionally, the sample size appears to be adequate for ensuring the statistical reliability of the results across the documents.

However, the reviewer has several questions and comments:

1. After excluding patients with confirmed COVID-19, the average age of the remaining patients shifted from 37 years to 22 years. This is a substantial difference, even if only older individuals were infected with COVID-19. Please re-evaluate the calculations to ensure no errors are present.

2. For improved clarity in Table 3, it would be beneficial to relocate the designation "n, (%)" to the headers of columns 2-4 and remove it from the first column.

3. The table titled " Interrupted time series analysis of respiratory virus circulation before, during, and after the COVID-19 pandemic in Jalisco, Mexico" is numbered as Table 3, even though Table 3 has already been assigned to " Respiratory virus detections other than SARS-CoV-2 in patients who presented with respiratory symptoms." There is an apparent error in the table numbering.

4. Throughout the text, the term "respiratory virus" is used in the singular form, while the study describes the multiple viruses. The term should be adjusted to "respiratory viruses" for consistency and accuracy.

Round 2

Reviewer 1 Report

Comments and Suggestions for Authors

Dear Authors,

Thank you for your comprehensive response to my comments. I have recommended that the study be accepted.